# Pre- Trained Language Models for Mental Health: An Empirical Study on Arabic Q&A Classification

**DOI:** 10.3390/healthcare13090985

**Published:** 2025-04-24

**Authors:** Hassan Alhuzali, Ashwag Alasmari

**Affiliations:** 1Department of Computer Science and Artificial Intelligence, Umm Al-Qura University, Makkah 24382, Saudi Arabia; 2Department of Computer Science, King Khalid University, Abha 62521, Saudi Arabia; aasmry@kku.edu.sa; 3Center for Artificial Intelligence (CAI), King Khalid University, Abha 62521, Saudi Arabia

**Keywords:** mental health, natural language processing, question/answer classification, text classification, pre-trained language models

## Abstract

**Background:** Pre-Trained Language Models hold significant promise for revolutionizing mental health care by delivering accessible and culturally sensitive resources. Despite this potential, their efficacy in mental health applications, particularly in the Arabic language, remains largely unexplored. To the best of our knowledge, comprehensive studies specifically evaluating the performance of PLMs on diverse Arabic mental health tasks are still scarce. This study aims to bridge this gap by evaluating the performance of pre-trained language models in classifying questions and answers within the mental health care domain. **Methods:** We used the MentalQA dataset, which comprises Arabic Questions and Answers interactions related to mental health. Our experiments involved four distinct learning strategies: traditional feature extraction, using PLMs as feature extractors, fine-tuning PLMs, and employing prompt-based techniques with models, such as GPT-3.5 and GPT-4 in zero-shot and few-shot learning scenarios. Arabic-specific PLMs, including AraBERT, CAMelBERT, and MARBERT, were evaluated. **Results:** Traditional feature-extraction methods paired with Support Vector Machines (SVM) showed competitive performance, but PLMs outperformed them due to their superior ability to capture semantic nuances. In particular, MARBERT achieved the highest performance, with Jaccard scores of 0.80 for the question classification and 0.86 for the answer classification. Further analysis revealed that fine-tuning PLMs enhances their performance, and the size of the training dataset plays a critical role in model effectiveness. Prompt-based techniques, particularly few-shot learning with GPT-3.5, demonstrated significant improvements, increasing the accuracy of question classification by 12% and the accuracy of answer classification by 45%. **Conclusions:** The study demonstrates the potential of PLMs and prompt-based approaches to provide mental health support to Arabic-speaking populations, providing valuable tools for individuals seeking assistance in this field. This research advances the understanding of PLMs in mental health care and emphasizes their potential to improve accessibility and effectiveness in Arabic-speaking contexts.

## 1. Introduction

Pre-Trained Language Models (PLMs) have made significant advances in recent years, revolutionizing applications in different domains, including medicine [1]. However, research dedicated to understanding and improving PLMs for mental health remains in its early stages. The application of PLMs in mental health has exciting possibilities for both patients seeking support and health care providers seeking to enhance their services. Patient applications offer a spectrum of experiences: immersive conversation directly with the model such as [2,3], or using PLMs to understand and categorize user input for connection with a human therapist such as [4]. For health care providers, PLMs can be used to generate recommendations or suggested responses.

However, the potential effectiveness of PLMs in the mental health domain depends on their ability to understand the nuances of human language, including the subjective and variable nature of mental health symptoms and the need for specialized skills. This becomes even more challenging when considering languages like Arabic. With more than 400 million native speakers worldwide, the Arabic language is a language of great richness and complexity [5]. Arabic is also among the top five spoken languages in the world. In addition, with over 280 million users worldwide, Arabic language ranks as the fourth most used language on the Internet [5].

Despite efforts worldwide in other languages [6,7,8], the Arabic language is an understudied language regarding mental health disorders. According to [9], there is a significant disparity in the availability of mental health datasets in different languages. The study reveals that English datasets dominate, accounting for 81% of the total, while Chinese datasets follow at 10%. In contrast, Arabic datasets are notably scarce, representing only 1.5% of the dataset resources. The scarcity of Arabic datasets can be attributed to several factors, including the inherent complexity of the Arabic language. Arabic is characterized by its rich morphology, diverse dialects, and a unique writing system that reads from right to left. Additionally, the absence of capitalization and the use of character shapes that vary based on their position within a word add to the challenges in data collection and annotation. Moreover, the high costs and efforts associated with creating high-quality annotated Arabic datasets further contribute to the limited availability of such resources.

To date, only a handful of works have considered mental health issues in the Arabic language [10,11,12,13,14]. In particular, Ref. [11] developed depression-detection models from Arabic texts on Twitter which focused on the cultural stigma surrounding depression in Arab societies. Another study by [12] also focused on the detection of depression in which they applied various machine learning algorithms and feature-extraction techniques.

### Research Objectives and Contributions

The primary goal of this study is to evaluate the effectiveness of PLMs for automatic classification of questions and answers in the Arabic mental health domain, using the newly released MentalQA dataset [15]. This classification task is a foundational step towards enabling future developments in automated mental health support systems, as it allows for the organization of user queries and expert responses into well-defined categories. By accurately classifying both questions and answers, the system can better understand the user’s intent and emotional state, enabling the generation of responses that are not only relevant to the specific type of mental health concern, but also aligned with the appropriate response strategy, be it offering emotional support, providing factual information, or recommending additional resources. In light of this objective, we present the following contributions:Conducting the first set of experiments on the MentalQA dataset, the novel Arabic mental health question-answering dataset featuring question-answering interactions.Exploring the effectiveness of Classical Machine Learning models and PLMs on the MentalQA dataset.Demonstrating the current capabilities and limitations of both classical models, Arabic PLMs and promoting GPT models for mental health care and ways for further improving the results of PLMs on the MentalQA dataset. By identifying the areas where these models excel and where they fall short, we can pave the way for further development and refinement.

This research will contribute valuable insights into the feasibility of using PLMs for Arabic mental health support, ultimately helping to create accessible and culturally sensitive resources for Arabic-speaking populations. The rest of this manuscript is organized as follows: Section 2 reviews relevant works in the field. Section 3 details the experiments conducted and the implementation details. Section 3.7 presents the results of our experiments, while Section 4 focuses on the evaluation of the results, examining factors that influence model performance. Section 5 discusses the results and implications, as well as highlighting some limitations and ethical considerations. Finally, Section 6 summarizes our work and proposes potential avenues for future research.

## 2. Related Work

Significant research efforts have been devoted to utilizing computational methods for mental health applications. This section delves into the relevant literature across three key domains. Firstly, we examine the established approaches of Classical Machine Learning (ML) techniques used for mental health tasks. Secondly, we explore the emerging field of PLMs and their potential for mental health applications. Finally, we analyze existing resources for mental health support.

### 2.1. Classical Machine Learning

A substantial body of research has explored computational methods for mental health applications. Classical Machine Learning (ML) techniques have been a fundamental of this endeavor, offering a diverse set of algorithms for various tasks. Several classical ML algorithms have demonstrated promising results in mental health applications, each with its own strengths and weaknesses. Techniques like Support Vector Machines (SVM) excel at classifying text data for tasks like depression detection, but lack interpretability [16]. Decision Trees offer a clearer view of decision-making but can become unstable with complex data [17]. Naïve Bayes is efficient but has limitations with intricate datasets. Despite their success, classical ML often requires extensive feature engineering, hindering their ability to handle unstructured text data. This paves the way for PLMs—the focus of the next section—which can automatically learn these features from vast amounts of data.

### 2.2. Pre-Trained Language Models

Significant progress has been made in developing PLMs for the English language, demonstrating their effectiveness in various Natural Language Processing (NLP) tasks. These powerful PLMs are trained on massive volumes of textual data, allowing them to perform a wide range of tasks. Prior models like ULMFiT [18] excel in text-classification tasks across languages, while BERT [19] established a foundation for “deep bidirectional pre-training”, achieving SOTA performance on various sentence-level tasks. Subsequent advancements addressed limitations of BERT. ALBERT tackles memory and training time issues with parameter-reduction techniques [20], while RoBERTa refines the pre-training process for improved performance [21]. XLNet goes further, addressing shortcomings in BERT’s handling of masked positions and pre-training approaches [22]. These advancements showcase the continuous development and growing capabilities of general domain PLMs for English NLP tasks.

Despite being the fourth most prevalent language online with more than 400 million speakers across 22 countries [5], only a few attempts have been made to construct PLMs in Arabic. This gap can be attributed to the inherent complexity of Arabic, characterized by its rich morphology, diverse dialects, and unique writing system that reads from right to left, lacks capitalization, and utilizes character shapes that vary based on their position within a word [23].

One of the first efforts to construct a pre-trained LLM specifically for Arabic was hULMonA [24], which is the first universal language model specifically designed for Arabic. This model obtained SOTA performance in Arabic sentiment analysis tasks, demonstrating its effectiveness on various Arabic datasets. AraBERT, another early effort, pre-trained the BERT model for Arabic, achieving advanced results in sentiment analysis, entity recognition, and question answering [25]. Subsequent studies explored multilingual models like GigaBERTs and focused on the impact of data sources like OSCAR and Gigaword [26]. ARBERT and MARBERT are two models that were built by [27], and they were trained on a large number of Modern Standard and Dialect Arabic datasets, respectively. Another model by [28] pre-train a single BERT-base model called CAMeLBERT, based on variations of dialect and classic Arabic data. Ref. [29] introduced “AlGhafa”, which is focused on benchmarking Arabic PLMs for multiple-choice evaluation. An additional work by [30] conducts benchmarking recent advancements in PLMS against state-of-the-art models on various NLP tasks.

### 2.3. Resources for Mental Health Support

While PLMs have been primarily focused on general domains, there is a growing interest in creating models tailored for specific domains like health. Researchers have developed several domain-specific PLMs for learning text representations in the health domain, such as BioMedical BERT [31] and Clinical BERT [32]. These models excel in capturing domain-specific nuances compared to general-purpose PLMs. However, PLMs designed specifically for mental health remain scarce. Pioneering efforts like MentalBERT and MentalRoBERTa address this gap, offering valuable resources for the mental health care research community [33]. Ref. [34] conducts a comprehensive review of existing work, highlighting the potential benefits and limitations of PLMs in early screening, digital interventions, and other clinical applications in mental health.

The efficacy of PLMs is demonstrably dependent upon the quality and quantity of data employed for their training. In the literature, efforts were made to develop benchmark corpora for the mental health support [35,36,37,38,39,40]. Existing examples include datasets for depression [37], self-harm [38], and stress [39]. However, the field is evolving towards more nuanced datasets that capture emotions related to specific mental conditions. The CEASE dataset focuses on emotions of suicide attempters [41], while EmoMent targets emotions linked to depression and anxiety [6]. Additionally, datasets are being developed for tasks like pain level identification in mental health notes [42] and causal interpretation using datasets like CAMS [43]. Moreover, a recent study focused on wellness classification and extraction from Reddit posts related to mental health [44]. To accomplish this, the authors began by constructing a new corpus specifically tailored for this task. Subsequently, they evaluated the effectiveness of PLMs in analyzing the corpus. These advancements highlight a shift towards richer and more comprehensive mental health data resources, crucial for training effective PLMs in this domain.

### 2.4. How Does This Research Differ from Existing Work?

While progress has been made, a significant gap exists in applying PLMs to Arabic language mental health applications. This gap stems from two main challenges. The first challenge revolves around the scarcity of Arabic language datasets, particularly those that are specifically tailored to the medical domain. The availability of such datasets is crucial for training PLMs to effectively comprehend and generate accurate content in the field of mental health. The second challenge lies in the limited availability of labeled data for mental health in the Arabic language. While the introduction of MentalQA, which is the first Arabic mental health dataset featuring question-answering interactions, has provided a valuable foundation, it is clear that more diverse and comprehensive datasets are required. Such datasets would enable the development of robust Arabic PLMs, capable of addressing a wider range of mental health applications. Given these limitations, we have been motivated to take action. Our objective is to benchmark the MentalQA dataset and utilize it as a stepping stone in supporting the development of powerful Arabic PLMs for mental health applications. Through this endeavor, we aim to bridge the existing gap and facilitate advancements in the field, ultimately benefiting individuals seeking mental health support in the Arabic-speaking world.

## 3. Experiments

### 3.1. Dataset Description

In this work, we selected the MentalQA dataset [15], which consists of posts written in Arabic. This dataset was compiled from an Arabic medical platform called Altibbi, focusing on mental health questions and answers posted between 2020 and 2021. The MentalQA dataset comprises of 500 questions posted by patients, followed by 500 corresponding answers provided by professional doctors. Questions are classified into seven types: diagnosis, treatment, anatomy, epidemiology, healthy lifestyle, provider choice and other. Additionally, answers are classified into three strategies: information, direct guidance, and emotional support. Table 1 further shows the complete range of categories that encompass both question and answer types within the dataset. To better illustrate the dynamics between patients and professional doctors, Figure 1 presents two annotated examples featuring different question and answer types. These examples serve as a visual representation of the interaction within the dataset and highlight the variety of exchanges that take place.

### 3.2. Task Setting

The MentalQA dataset encompasses two tasks: the classification of question types and answer types. Both tasks allow for the assignment of multiple labels, employing a multi-label classification approach. To ensure proper evaluation, we divided the dataset into three subsets: a training set, accounting for 60% of the total data, a validation set, representing 20% of the total data, and a test set, also encompassing 20% of the total data. To gain a thorough understanding of the MentalQA dataset, Table 2 provides a comprehensive overview of the dataset. This table includes information about the number of instances in the training, validation, and test sets, as well as the different types of questions and answers available in this dataset.

### 3.3. Experimental Design

In this paper, we explore four different approaches for the task of Q/A multi-label classification: traditional feature-extraction methods with Support Vector Machines (SVM), utilizing PLMs as feature extractors with SVM, fine-tuning PLMs, and prompting engineering. Our goal is to assess the performance of these approaches specifically for the task of the MentalQA dataset described in the above section. In addition, we include two common baselines that are based on the most frequent class and randomness. Figure 2 provides an illustration of our experimental design.

Traditional feature-extraction methods have been widely used in text classification [45,46,47]. These methods involve converting input text into numerical representations based on word or concept frequency or presence. Techniques like “term frequency–inverse document frequency *(TF-IDF)*” and counting methods have been extensively employed. These approaches offer simplicity and interpretability, relying on straightforward calculations. Subsequently, SVM, a popular machine learning algorithm, is employed to classify the transformed text data. The goal of SVM is to seek out an optimal hyperplane within the feature space, effectively segregating distinct classes. The effectiveness of this approach has been demonstrated in various studies [48].

The emergence of PLMs has revolutionized NLP tasks, including text classification [19,20,21]. These models capture contextualized representations of words and sentences by leveraging large-scale unsupervised training on vast amounts of text data. Instead of relying on explicit feature-extraction methods, we utilize PLMs as feature extractors. These models encode the input text into dense vector representations, capturing intricate semantic and syntactic information. The resulting representations are then fed into an SVM classifier, which learns to discriminate between different classes. This approach has shown remarkable performance improvements in various NLP applications and has been validated in previous research [45,49].

Building upon the concept of using PLMs, we consider the approach of fine-tuning these models. Fine-tuning involves training the PLMs on a task-specific dataset, allowing them to adapt to the specific classification objective. By fine-tuning PLMs, our aim is to capture task-specific nuances and optimize the models for improved classification performance. This approach has demonstrated promising results, as supported by research on universal language model fine-tuning [18,19,50,51]. Fine-tuning enables the models to effectively leverage their contextualized representations for accurate predictions.

In the realm of Artificial Intelligence (AI), the role of prompting engineering is pivotal in steering AI models towards producing outputs that align with user intentions. By providing specific input or instructions, the AI model can generate responses that are relevant, coherent, and in line with desired outcomes. This research paper focuses on harnessing the capabilities of GPT-3.5 and GPT-4. We closely examine the prompt used to explore the potential of these GPT models. The prompt provides instructions for the model to classify the text based on a predefined set of categories. To illustrate this process, here s an example:Task [task prompt]: Your task is to analyze the question in the TEXT below from a patient and classify it with one or multiple of the seven labels in the following list: “Diagnosis”, “Treatment”, etc.Text [input text]: Be persistent with your doctor, and inform them of your concerns. Embrace the spiritual aspects and try to nurture them, as they will assist you in transcendental thinking and relaxation.Response [output]: Guidance, emotional support.

However, the zero-shot setting often resulted in inconsistent or inaccurate outputs, particularly when the input was ambiguous, lacked direct cues, or involved multiple overlapping categories. These limitations were most evident in cases where subtle contextual clues were necessary to determine the appropriate classification. To address this, we employed a few-shot prompting strategy, where the model was presented with a small number of labeled examples (2–3) before processing the target input. Although the structure of the task prompt remained similar to that in the zero-shot case, the inclusion of representative examples helped the model better infer the classification logic. This technique enhanced the model’s ability to identify relevant labels and improved its overall performance by reducing misclassifications and promoting consistency across outputs. In addition, we kept the model’s hyperparameters, including temperature, top-p, and max tokens, at their default settings across all experiments to maintain consistency and reproducibility. Our focus was solely on evaluating the impact of prompt design rather than tuning generation parameters.

### 3.4. Application of Learning Approaches in MentalQA

To clarify the application of each learning approach within the MentalQA: For the traditional feature-based models, we extracted textual features such as TF-IDF vectors from the MentalQA dataset. These representations were then used to train Classical Machine Learning classifiers, including SVM, to predict one or more category labels for each question or answer. In the PLMs as feature extractors approach, we utilized pretrained Arabic language models to encode each MentalQA instance into contextual embeddings. Specifically, we extracted the representation of the CLS token, which was then passed to an SVM classifier trained on the MentalQA labels. For the fine-tuned PLMs, we extended the pretrained models by adding a task-specific classification head and fine-tuned the entire model end-to-end using the labeled MentalQA training data. This approach allowed the model to adapt more directly to the language and label distribution specific to our mental health-classification task. Finally, for the prompt-based approaches using GPT-3.5 and GPT-4, we provided MentalQA examples as natural language prompts. In the zero-shot setting, the model was given only task instructions and a target sample. In the few-shot setting, we included 2–3 representative MentalQA examples in the prompt to guide the model’s classification behavior. In both cases, the model outputs were parsed and evaluated against ground truth labels using the same metrics as the supervised approaches. Each learning strategy was applied consistently using the same data splits and evaluation procedures to ensure fair comparison across methods.

### 3.5. Training Objective

In the context of multi-label classification, the task involves selecting one or multiple labels. To determine whether $P_{i}$ represents the correct question/answer type, we use the Sigmoid activation function, and the binary cross entropy loss (BCE) when calculating the classification loss. The formalization of this loss function is as follows:(1)$$\begin{array}{c} Loss_{BCE}=\frac{1}{\left|L\right|}\sum_{i=1}^{\left|L\right|}\left(Y_{ti}\log\left(P_{li}\right)+\left(1-Y_{ti}\right)\log\left(1-P_{li}\right)\right) \end{array}$$
where $\left|L\right|$ denotes the number of either question or answer types, depending on the specific task at hand. $Y_{t}$ corresponds to the ground truth label, while $P_{li}$ refers to the probability associated with the *i*-th label.

### 3.6. Implementation Details

For our experiments, we utilized PyTorch [52] as our framework of choice. The experiments were performed using a T4-GPU equipped with 15 GB of memory. For both extracting features from PLMs and fine-tuning them, we made use of the open-source Hugging-Face implementation [53]. We selected only three Arabic PLMs due to resource constrains: i.e., “AraBERT” developed by [25], “MARBERT” developed by [27], and “CAMeLBERT-DA” developed by [28]. According to the study conducted by [34], both “AraBERT” and “MARBERT” demonstrate strong performance in detecting depression. Therefore, they are suitable candidates to be used in our benchmark study. For prompt engineering results, we used the OpenAI API to conduct our experiments considering two variants, i.e., GPT-3.5 and GPT-4.

For evaluation, given that the two tasks involve multi-label classification, we employed three widely used metrics [50,54]: i.e., Micro F1-score, weighted F1-score, and Jaccard index score. The presence of a class imbalance issue in the MentalQA dataset is worth mentioning. To ensure a more equitable evaluation of the model’s performance, we opt for the weighted-F1 score instead of macro-F1, as it takes into account the varying class distribution and mitigates the effects of imbalanced classes.

To ensure consistency, all models used in this paper were trained using identical hyperparameters and a fixed initialization seed. The hyperparameter settings consisted of a feature dimension of 786, a batch size of 8, a dropout rate of 0.1, an early stop patience of 10, and a training duration of 15 epochs. For optimization, we selected Adam optimizer [55] with a learning rate of $2\times 10^{-5}$. Furthermore, we leveraged scikit-learn [56] as a valuable resource for implementing the SVM algorithm as well as extracting both counting-based and TF-IDF features. It is important to note that all aforementioned models were tuned exclusively on the validation set. Table 3 provides a summary of the aforementioned hyperparameters.

### 3.7. Experimental Results

Table 4 (this study focused on exploring the MentalQA dataset and it conducted the first set of experiments on this particular dataset; as a result, there are currently no existing SOTA models available for comparison) shows the performance of the three approaches explored on the tasks of classifying question and answer types. The reason for selecting these approaches is to benchmark the MentalQA dataset against both Classical Machine Learning models and PLMs, which are widely used in both the NLP and mental health domains. This comparison allows us to explore the relative effectiveness of traditional approaches versus more recent advancements in language models. The evaluation metrics used to assess the performance include micro F1-score, weighted F1-score, and Jaccard score. We selected these evaluation metrics due to their broad applicability in multi-label classification tasks [50,54]. These metrics can be described as follows:F1-Micro aggregates the contributions of all classes to compute the average performance, which is useful when all classes are equally important.F1-Weighted accounts for label imbalance by weighting the F1 score of each class by its support (i.e., number of true instances).Jaccard Score (also known as Intersection over Union) is particularly suitable for multi-label settings, as it measures the similarity between the predicted and true label sets.

Regarding the evaluation methodology, we follow standard practices commonly used in multi-label classification tasks. Our model outputs a set of predicted labels for each input instance, which is then compared against the corresponding set of ground-truth labels. A data point is considered correct when its predicted label set matches the true label set, either fully (for exact match metrics) or partially, depending on the metric used (e.g., Jaccard allows partial overlaps). For threshold-based models, we used a fixed threshold (e.g., 0.5) to determine whether a label is assigned. If the model assigns a confidence score ≥ 0.5 to a particular label, that label is included in the predicted set.

We also compared the results against the following baselines, i.e., random and the most frequent class. The random baseline approach serves as a simple benchmark by randomly assigning question and answer types, resulting in low performance across all evaluation metrics. This highlights the need for more sophisticated models in question and answer classification. The most frequent class baseline approach selects the most common question and answer types as predictions for all samples, outperforming the random baseline but still falling short compared to more advanced models.

The first explored approach involves traditional feature extractor methods utilizing SVM with different feature representations. These models leverage occurrence-based, count-based and TF-IDF representations to capture word presence/absence, word frequency and importance in the input data, respectively. The SVM classifier utilizes these features to achieve higher performance scores compared to the baselines. The TF-IDF representation generally outperforms the occurrence-based and count-based representation, indicating that weighting words based on importance enhances model performance. Furthermore, it is worth noting that the SVM with TF-IDF features exhibited competitive performance to PLMs, as supported by previous research [48]. This outcome is to be expected, considering the limited size of the training set. Nevertheless, it is important to acknowledge that this situation might alter if the dataset expands to a larger scale.

The second approach utilizes PLMs as feature extractors in combination with the SVM classifier. AraBERT, CAMelBERT, and MARBERT, three Arabic PLMs, are employed to extract contextualized representations of the input data. These models achieve competitive performance scores for both question and answer types, benefiting from their ability to capture the semantic meaning of the text. MARBERT outperformed both Arabic PLMs in almost all setups and metrics.

The third approach fine-tuned PLMs, which takes it a step further by adapting the PLMs specifically to the tasks of question and answer classification. Through fine-tuning the three Arabic PLMs, these models achieve even higher performance scores compared to using the PLMs solely as feature extractors. Fine-tuning allows the models to learn task-specific patterns and optimize their performance on the given dataset. The results of this approach demonstrates that Fine-tuning MARBERT achieved the best performance.

The fourth approach explores the use of prompting PLMs such as GPT-3.5 and GPT-4. The objective was to evaluate their performance in two different scenarios: zero-shot learning and few-shot learning, specifically focusing on question and answer classification. The results indicate that the few-shot learning setting, utilizing only three instances of labeled data, yields better outcomes compared to the zero-shot learning setting. Across all three metrics, there was an overall performance improvement of up to 7%. This finding suggests that few-shot learning holds great potential for future research aimed at enhancing model performance while working with limited labeled data, which is often the case in real-world applications. By leveraging this approach, it becomes possible to achieve notable advancements even with a small amount of available data.

The performance of the four discussed approaches for classifying question and answer types was evaluated and compared to simple baselines. The first approach utilized traditional feature extractors with SVM, achieving higher performance than the baselines by capturing word frequency and importance through count-based and TF-IDF representations. The second approach employed PLMs as feature extractors, with AraBERT, CAMelBERT, and MARBERT achieving competitive scores by capturing semantic meaning. Fine-tuning the PLMs further improved performance by adapting them specifically to the question- and answer-classification tasks, resulting in even higher scores. The fourth approach involves promoting GPT models in two different settings, where the second one exhibited a strong performance. The findings from these results demonstrate the usefulness of each model employed and the utilization of contextualized representations for achieving accurate classification of question and answer types. These results also emphasize the crucial role played by the selected models and the value of incorporating contextual information in effectively categorizing question and answer types. The results obtained from GPT indicate that few-shot learning has the potential to substantially enhance model performance, even when working with a limited number of labeled data instances.

## 4. Evaluation of the Results

In this section, we conducted a series of comprehensive analyses that contribute to a deeper understanding of our experiments undertaken in this paper. These analyses encompassed various aspects and aimed to shed light on key factors influencing model performance. In the first analysis, our focus centered on evaluating the effect of fine-tuning PLMs compared to not fine-tuning them. By exploring both scenarios, we gained crucial insights into the benefits and trade-offs associated with fine-tuning PLMs. This analysis enabled us to make informed decisions regarding the optimal approach for our specific task. The second analysis shifted towards investigating the influence of data size on the performance. We assessed the influence of data availability on the performance of the models by systematically adjusting the size of the training set. This examination provided valuable insights into the relationship between data quantity and model performance. Furthermore, we conducted a detailed case study as part of the third analysis. The goal of this investigation was to identify potential errors or limitations in the employed models. By examining their performance and scrutinizing any discrepancies or inaccuracies, we aimed to identify areas for improvement and guide future work in addressing these issues effectively.

### 4.1. Effect of Fine-Tuning PLMs

In this section, we conducted an analysis to assess the effect of fine-tuning PLMs compared to not fine-tuning them. The results presented in Figure 3 focus on the performance of MARBERT. It was observed that this models achieved very low performance when it was not fine-tuned using the train set of the MentalQA dataset. This finding is consistent with prior research, which indicates that PLMs are typically trained on general domain data [57]. Consequently, fine-tuning becomes an essential step in order to achieve good performance, even with a small amount of labeled data. However, fine-tuning MARBERT exhibited the best performance. These results clearly indicate that fine-tuning the models significantly improves its performance in both question- and answering-type-classification tasks, specifically, when trained on task-specific data [18,19,58]. We now turn to discussing the effect of fine-tuning the same model using varying data sizes.

### 4.2. Effect of Zero-Shot vs. Few-Shot Learning

In this analysis, our objective was to assess the effectiveness of prompting PLMs like GPT as shown in Figure 4. To achieve this, we chose two prominent pre-trained language models, namely GPT-3.5 and GPT-4, and evaluated their performance in two specific learning settings: zero-shot learning and few-shot learning, utilizing just three labeled examples. Surprisingly, the results revealed that GPT-3.5 outperformed GPT-4 in both scenarios. Notably, even with the limited labeled data, GPT-3.5 showcased superior performance, showcasing its ability to leverage small labeled datasets effectively. These findings suggest that GPT-3.5 exhibits a strong capacity for understanding and generalizing from limited instances, surpassing the advancements made in GPT-4. One possible explanation is that GPT-4’s broader contextual reasoning capabilities may have led it to overgeneralize or reinterpret ambiguous inputs in ways that deviated from the classification task. For instance, in one case, GPT-4 labeled a question as “Diagnosis” while GPT-3.5 correctly labeled it as “Diagnosis & treatment” based on implicit cues. This highlights the importance of prompt alignment and model interpretability when deploying PLMs in sensitive domains like mental health.

In the zero-shot setting, GPT-3.5 demonstrated a micro F1-score of 0.59, a weighted F1-score of 0.55, and a Jaccard score of 0.45 for question classification, indicating its ability to generate reasonably accurate predictions without any specific training on the given task. However, when transitioning to the few-shot setting with only three labeled examples, GPT-3.5 experienced a noticeable increase in performance. It achieved a micro F1-score of 0.66, a weighted F1-score of 0.61, and a Jaccard score of 0.53. This improvement suggests that even with a minimal amount of labeled data, GPT-3.5 was able to leverage the provided examples and extract meaningful patterns, resulting in higher precision and recall rates. The significant boost in performance highlights the model’s capacity to adapt and utilize limited supervised information effectively, showcasing its ability to generalize and make more accurate predictions in the few-shot learning scenario. The benefits of few-shot learning in boosting performance were also observed in the context of answer classification.

It should be noted that, while GPT models in the zero-shot setting have demonstrated impressive performance across a wide range of general NLP tasks, their performance in classifying answer types within the domain of Arabic mental health was notably limited. This underperformance may be attributed to the absence of domain-specific fine-tuning, as GPT models are typically pre-trained on large-scale, diverse corpora that lack sufficient coverage of mental health discourse—particularly in Arabic. The nuanced nature of therapeutic interactions, coupled with the linguistic and cultural intricacies of Arabic, likely contributes further to this gap in performance. These findings underscore the limitations of zero-shot learning when applied to specialized, sensitive domains such as mental health, and point to the need for targeted domain adaptation and fine-tuning of large language models to ensure better accuracy and contextual understanding in future work.

### 4.3. Effect of Data Size on Performance

The analysis conducted in this section involves evaluating the performance of the MARBERT model (we chose this model because it achieved almost the best performance on all metrics) for question type and answer type classification. The study examines the impact of varying the size of the training data on the model’s performance. The evaluation utilizes three metrics, i.e., F1-Micro, F1-Macro, and Jaccard-Score, to assess the model’s performance. The analysis is presented in Figure 5 through two plots, where the top plot represents question type classification and the bottom plot represents answer types classification.

Figure 5 displays the influence of data size on model performance. The x-axis in both plots represents the size of the data, ranging from 50 samples to 200 samples, while the y-axis represents the corresponding performance scores. The analysis reveals a clear relationship between data size and model performance. The results obtained from the plots indicate that increasing the data size positively affects the model’s performance in both question and answer type classification. Notably, as the data size increases, there is a substantial improvement in performance. For question type classification, the F1-Micro score rises from 0.65 to 0.80, the F1-Macro score increases from 0.57 to 0.77, and the Jaccard score improves from 0.55 to 0.73. Similarly, the results of answer type classification display a similar trend, confirming the positive impact of larger datasets on model performance.

The analysis results emphasize the importance of data size in attaining improved performance scores for both question- and answer-type-classification tasks. These findings demonstrate the effectiveness of harnessing data size as a means to enhance the model performance. By increasing the amount of available data, the model’s performance can be substantially enhanced, leading to more accurate results in both question and answer type classification tasks.

### 4.4. Case Study

Upon analyzing the model predictions, we identified several noteworthy issues as shown in Table 5. Firstly, the model’s inability to accurately predict the emotional support label in the first three instances highlights its limitations in capturing the intricate emotional nuances within the text. Extensive research has emphasized the significance of emotions in comprehending and addressing mental health concerns [59,60]. The model’s errors in recognizing emotions indicate the need for further improvement in this aspect, as accurate emotional understanding is crucial for providing appropriate guidance.

Furthermore, the last three example sheds light on a common challenge in multi-label classification, which involves defining clear boundaries between certain labels. As mentioned in the works of [50,61], the relationships between labels directly impact the performance of the model. Addressing this challenge and considering label relationships could potentially enhance the model’s performance. In numerous cases, the labels “diagnosis and treatment” are frequently encountered. Exploring the dynamics between labels and developing strategies to handle them effectively in multi-label question classification could serve as an interesting area for future research and improvement.

## 5. Discussion

Our experiments yielded promising results, highlighting the potential of fine-tuned Arabic PLMs for Arabic mental health support systems. Fine-tuned models, like MARBERT, consistently outperformed traditional feature-extraction methods with SVM for both question- and answer-classification tasks and that is evident in the prior work [29,30]. This margin of improvement suggests that PLMs are adept at capturing the nuances and context of Arabic language specific to the domain of mental health. Notably, fine-tuning the PLMs on a dataset curated specifically for mental health questions and answers resulted in a significant performance boost compared to the non-fine-tuned model. This finding underscores the importance of domain adaptation—tailoring the PLMs to the specific domain through targeted training on relevant data. Additionally, our exploration revealed a clear correlation between dataset size and model performance, echoing prior research findings in NLP that larger datasets consistently yield better results [62]. This highlights the crucial role of data availability in this field, where the under-representation of Arabic mental health resources necessitates further data-collection efforts.

In addition, our study also sheds light on an interesting dynamic concerning general domain Arabic PLMs (AraBERT, CAMelBERT, MARBERT) when applied to specific tasks. While these models achieved promising performance on the MentalQA dataset, they might benefit from further training on even more relevant mental health corpora. This highlights the potential for developing Arabic PLMs specifically trained on mental health data, like a “MentalBERT” or “MentalRoBERTa” model [33]. Such models could be pre-trained on a massive dataset of Arabic text specifically tailored to the mental health domain, including mental health questions, clinical notes, support group discussions, and mental health resources. This targeted training could allow the PLMs to develop a deeper understanding of the nuances and terminology used in mental health conversations, potentially leading to even better performance on downstream classification tasks within this domain.

The results from the fine-tuned models on the mental health dataset lays the groundwork for exploring the effectiveness of such domain-specific Arabic PLMs for mental health applications. This aligns with calls for further research on the application of NLP for mental health tasks, particularly in under-resourced languages like Arabic [9]. Future research can investigate how these models can be further optimized to achieve specific goals, such as improved accuracy in type of questions sought or enhanced ability to identify users at risk. Additionally, exploring techniques like semi-supervised learning or leveraging unlabeled data through methods like contrastive learning could potentially enhance model performance while mitigating the limitations of dataset size. By pursuing these research directions, we can contribute to the development of more robust and effective Arabic language models for mental health support systems, ultimately improving accessibility to mental health resources for Arabic-speaking communities.

### 5.1. Implications

Our study using the MentalQA dataset demonstrates the potential of PLMs for Arabic mental health support systems, particularly in question- and answer-classification tasks. These findings hold significant implications for the future of mental health resources in general and more specifically in Arabic. By effectively classifying questions and answers, PLMs can facilitate intervention assistance by directing users to appropriate resources or connecting them with potential therapists based on their specific needs. This targeted approach can significantly reduce the time and effort required for individuals seeking help.

In addition to improving the screening processes, the findings also have important implications for accessibility in Arabic-speaking communities. Many individuals in these communities face multiple barriers to mental health care, including geographic isolation, socioeconomic constraints, and social stigma surrounding mental health issues. By enabling more effective processing of Arabic language content, PLMs have the potential to support the development of culturally sensitive, linguistically appropriate digital tools. These tools could serve as a bridge to care, providing individuals with non-judgmental, easily accessible, and private avenues to seek mental health information and support, a crucial step in addressing the widespread treatment gap in mental health care.

Another important implication relates to cultural and linguistic sensitivity. The results highlight that PLMs trained or fine-tuned on Arabic data perform better than general-purpose models, demonstrating the importance of grounding technological solutions in the cultural and linguistic realities of the target population. In mental health contexts, where language is used to express highly personal and culturally nuanced experiences, ensuring that PLMs can correctly interpret such expressions is essential for providing meaningful support. This demonstrates the need to invest more in developing domain-specific and culturally aware PLMs. Such models are essential to capture the unique ways mental health concerns are discussed in Arabic-speaking contexts.

Furthermore, the success of prompt-based techniques—particularly few-shot learning with GPT-3.5—highlights the potential for rapid deployment of these models in real-world systems, even in the absence of extensive training data. This is particularly valuable in resource-constrained health care systems, where annotated data for mental health conversations may be scarce. By lowering the technical and financial barriers to developing effective classification models, these approaches open the door for a broader range of health care providers, non-profits, and community organizations to deploy digital mental health tools tailored to the needs of Arabic-speaking populations.

While this study focused on fine-tuning pretrained language models to optimize classification performance, we acknowledge that this approach can be computationally demanding. Few-shot prompting offers a more resource-efficient alternative, particularly in scenarios with limited computational capacity. Although a direct cost-benefit comparison was beyond the scope of this work, we recognize the value of such analysis and consider it a promising direction for future research.

We also considered whether the GPT models exhibited biases toward specific classification categories. While our results did not reveal strong or systematic biases, we observed that both GPT-3.5 and GPT-4 occasionally failed to recognize emotional support-related cues, particularly when such cues were subtle or implied rather than explicitly stated. This limitation is especially important in the context of mental health support, where the recognition of emotional needs is critical. Future research should include a more detailed bias analysis and explore strategies to improve the detection of nuanced emotional content in multilingual and culturally diverse contexts.

Finally, while the study demonstrates clear promise, it is important to recognize that these technologies are intended to enhance, not replace, professional mental health care. Automated classification and response systems can play a valuable supporting role in scaling up services, but human oversight remains essential, particularly in cases involving high-risk individuals or complex psychological issues. Future research should explore not only the technical performance of these models, but also their clinical impact, user acceptability, and integration into existing mental health care pathways.

### 5.2. Limitations and Ethical Considerations

In light of our experimental findings on the MentalQA dataset, we have obtained promising results in discerning question and answer types. However, it is important to acknowledge that our study encompasses certain potential limitations and highlights avenues for future exploration. Our experiments were not designed for diagnostic purposes, but rather aimed at offering an estimation for categorizing Q/A types, thereby facilitating intervention assistance and providing guidance for non-clinical applications.

Firstly, a primary limitation of this study is the size of the MentalQA dataset, which comprises 500 question-answer pairs. While the MentalQA dataset offers a crucial initial resource for Arabic mental health NLP, its limited size necessitates careful interpretation of our results. To address this inherent constraint, future research should prioritize the development and curation of larger and more diverse Arabic mental health datasets. In addition, exploring data augmentation techniques tailored to the nuances of the Arabic language and the specific characteristics of mental health-related text, as well as investigating semi-supervised learning methodologies to leverage potentially abundant unlabeled Arabic text data, could be beneficial avenues to mitigate the limitations of the current dataset size.

Secondly, our reliance on automated classification, even with fine-tuned PLMs, carries inherent risks. Misclassification of user queries or expert responses could lead to the provision of irrelevant or even unhelpful information, potentially causing distress or hindering access to appropriate support. The subjective nature of mental health experiences and the subtle nuances in language that indicate emotional states pose a significant challenge for even the most advanced models.

Furthermore, ethical considerations surrounding data privacy and security are paramount. The MentalQA dataset, while anonymized, originates from sensitive user interactions related to mental health. Future work involving larger and more diverse datasets must prioritize robust anonymization techniques and adhere to strict ethical guidelines to protect user confidentiality and prevent potential misuse of sensitive information.

Finally, the potential for bias within the Pre-Trained Language Models themselves is a crucial concern. These models are trained on vast amounts of text data, which may reflect societal biases related to mental health, cultural norms, and linguistic variations. Such biases could lead to discriminatory outcomes or the perpetuation of harmful stereotypes in automated mental health support systems. Therefore, ongoing efforts are needed to identify and mitigate these biases to ensure fairness and equity in the application of PLMs in this sensitive domain.

## 6. Conclusions

In conclusion, this study explored the effectiveness of Classical Machine Learning models and Pre-Trained Language Models (PLMs) on the MentalQA dataset, the Arabic mental health question-answering dataset featuring question-answering interactions. While traditional feature extractors with SVM achieved a strong performance, PLMs like MARBERT and AraBERT demonstrated even better performance due to their ability to capture semantic meaning. These results suggest that PLMs hold significant promise for Arabic mental health question-answering, providing intervention services or connecting patients with resources. Furthermore, the experiment of prompt engineering has provided valuable insights into the advantages of few-shot learning when compared to zero-shot learning. Through our exploration, we were able to enhance question-classification results by 12% and improve answer-classification results by 45%. These findings demonstrated the potential of PLMs in the domain of Q&A classification for mental health care. By leveraging PLMs, we can facilitate the development of accessible and culturally sensitive resources specifically tailored to Arabic-speaking populations.

This, in turn, can help overcome barriers related to accessibility and stigma by enabling culturally-sensitive, language-aware support in low-resource settings. Our findings highlight the importance of tailoring PLMs to the cultural and linguistic context in which they are deployed, ensuring that technology is both effective and culturally competent. As digital mental health services continue to expand, these advances can play a critical role in improving mental health outcomes for Arabic-speaking individuals, particularly in regions with limited access to professional care.

## Figures and Tables

**Figure 1 healthcare-13-00985-f001:**
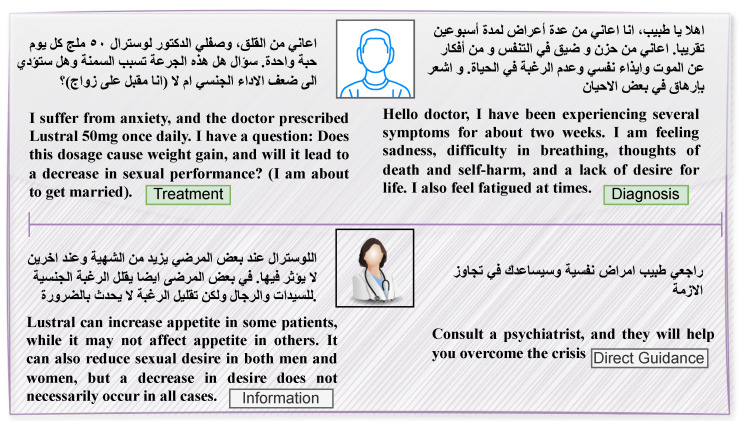
Example of two annotated Q&A posts in MentalQA dataset, with each Q&A post translated into English for better readability. The first row represents the questions, while the second row represents the corresponding answers. Additionally, the categories for each question and answer are included.

**Figure 2 healthcare-13-00985-f002:**
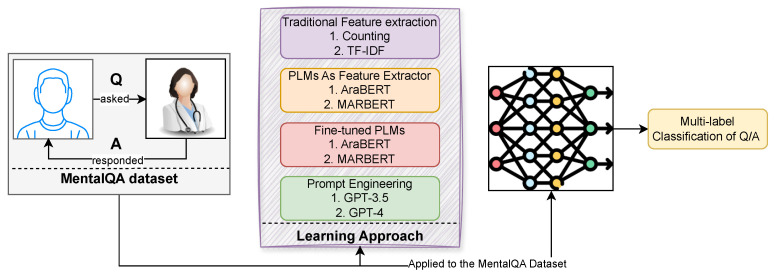
An overview of our experimental design. Specifically, it outlines the process by which the input is conveyed to the design of the learning approach, wherein the resulting outputs of various approaches are linked to the desired task outcome, namely, a multi-label classification of Q/A types.

**Figure 3 healthcare-13-00985-f003:**
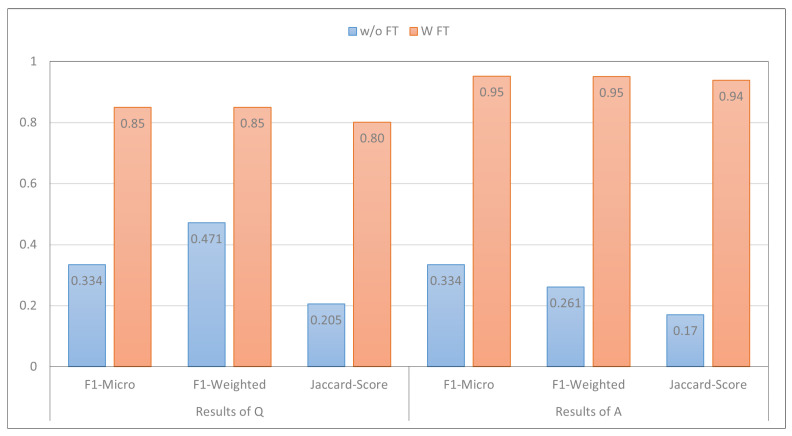
The impact of fine-tuning PLMs compared to not fine-tuning them. The x-axis of the plot represents the metrics employed in the paper, while the y-axis represents the corresponding model scores. Additionally, the color bars within the plot indicate with fine-tuning (w/ FT) vs. without fine-tuning (w/o FT).

**Figure 4 healthcare-13-00985-f004:**
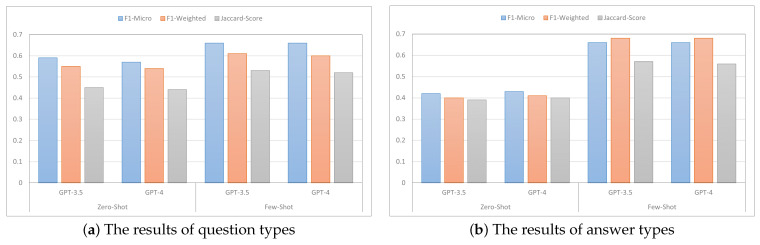
Both plots depict the impact of few-shot learning on model performance.

**Figure 5 healthcare-13-00985-f005:**
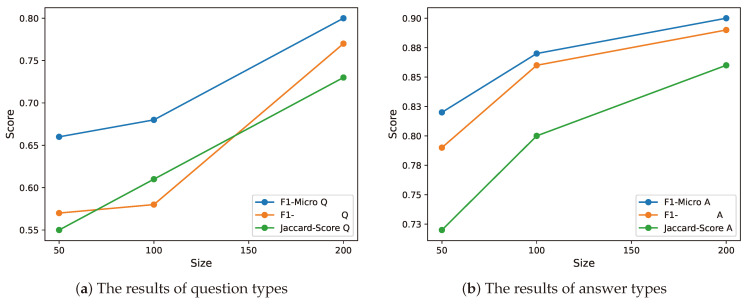
Both plots depict the impact of data size on model performance. The x-axis in the plot indicates the number of samples utilized for training, while the y-axis corresponds to the score of each metric.

**Table 1 healthcare-13-00985-t001:** Question (Q) and Answer (A) types.

#	Q-Types	#	A-Types
1	Diagnosis	1	Information
2	Treatment	2	Direct Guidance
3	Anatomy	3	Emotional Support
4	Epidemiology		
5	Healthy Lifestyle		
6	Health Provider Choice		
7	Other		

**Table 2 healthcare-13-00985-t002:** Data statistics.

Info./Task	Q-Types	A-Types
Train (#)	300	300
Validation (#)	100	100
Test (#)	100	100
Total (#)	500	500
Classes (#)	7	3
Setup	multi-label	multi-label

**Table 3 healthcare-13-00985-t003:** A summary of hyperparameters utilized in this work.

Parameter	Value
dimension	768
Batch-size	8
Dropout	0.1
Early-stop-patience	10
#epochs	15
learning rate	$2\times 10^{-5}$
Optimizer	Adam

**Table 4 healthcare-13-00985-t004:** The results of multi-label Q/A type classification on the test set of MentalQA. “Z-Shot” refers to “zero-shot”, whereas “F-Shot” refers to “few-shot”. Best: **bold**.

	Results of Q Classification			Results of A Classification		
Model	F1-Micro	F1-Weighted	Jaccard-Score	F1-Micro	F1-Weighted	Jaccard-Score
**Baseline**						
Random	0.33	0.26	0.21	0.50	0.45	0.34
Most common class	0.62	0.75	0.49	0.79	0.83	0.67
**SVM**						
SVM (Occurrences)	0.83	0.84	0.78	0.91	0.92	0.87
SVM (Count)	0.83	0.84	0.78	0.93	0.93	0.90
SVM (TF-IDF)	0.84	0.85	0.79	0.93	0.93	0.90
**PLMs As Feature Extractor**						
SVM (AraBERT)	0.80	0.81	0.74	0.92	0.92	0.89
SVM (CAMelBERT-DA)	0.81	0.81	0.79	0.92	0.93	0.91
SVM (MARBERT)	0.81	0.81	0.79	0.92	0.92	0.89
**Fine-tuning PLMs**						
AraBERT	0.84	0.82	0.79	0.94	0.94	0.92
CAMelBERT-DA	**0.85**	0.83	**0.80**	0.94	0.93	0.92
MARBERT	**0.85**	**0.85**	**0.80**	**0.95**	**0.95**	**0.94**
**Prompt Engineering**						
GPT-3.5 (Z-Shot)	0.59	0.55	0.45	0.42	0.40	0.39
GPT-4 (Z-Shot)	0.57	0.54	0.44	0.43	0.41	0.40
GPT-3.5 (F-Shot-3)	0.66	0.61	0.53	0.66	0.68	0.57
GPT-4 (F-Shot-3)	0.66	0.60	0.52	0.66	0.68	0.56

**Table 5 healthcare-13-00985-t005:** Comparing model’s predictions to the actual labels, specifically in the context of answer and question classification. The terms “info”, “guid”, “emo”, “diag”, “treat”, “prov”, and “H.life” are abbreviations that stand for information, guidance, emotional support, diagnosis, treatment, health provide choice, and health lifestyle, respectively. The translated text is the result of utilizing GPT3.5-Turbo to translate the original text from Arabic to English. We excluded the original text for the sake of space.

#	Translated Text	Actual	Prediction
Examples representing Answers			
1	Be persistent with your doctor, and inform them of your concerns. Embrace the spiritual aspects and try to nurture them, as they will assist you in transcendental thinking and relaxation.	Guid, Emo	Guid
2	Good evening. It is necessary to consult a mental health professional to assess your condition. In the meantime, think positively about yourself and reconcile with yourself, loving yourself.	Guid, Emo	Guid
3	The sadness will end, and you will live in complete happiness, God willing. I advise you to speak to a specialist in psychological therapy, as there are modern methods and wonderful medications that can help you, God willing, to live happily. Many people go through tough times and overcome them, and you are one of them. Do not suppress your sadness, and consult someone you trust.	Guid, Emo	Guid
Examples representing Questions			
4	I went to a psychiatrist two years ago and received treatment there. The symptoms disappeared, and I felt much better, so I stopped the treatment. However, now the sleep disturbances and anxiety have returned. What should I do? Should I go back to the doctor?	diag, treat, prov	treat
5	“I am a 40-year-old man who has never been married before, but now I am considering getting married. However, some people advise me against getting married at this age, saying that I won’t be able to live a happy life after reaching this age”.	H.life	diag, treat
6	The withdrawal symptoms of escitalopram are intense for me when I stopped taking it due to fear of its impact on pregnancy. However, the symptoms were strong, and I visited the doctor feeling unwell. My blood tests and urine tests came out normal, but I experienced a setback after discontinuing the medication two months ago. I am unable to visit the psychiatrist, and I have been using a dose of 10mg and then 20mg for approximately two and a half years. I need urgent help.	diag, treat	treat

## Data Availability

The data presented in this study are openly available in https://github.com/hasanhuz/MentalQA.

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
