# Peer review of "Pre- Trained Language Models for Mental Health: An Empirical Study on Arabic Q&A Classification"

_healthcare, 2025, doi:10.3390/healthcare13090985_

Round 1

Reviewer 1 Report

Comments and Suggestions for Authors

This paper makes a valuable contribution to Arabic NLP for mental health applications. However, improvements in dataset discussion, statistical analysis, prompt engineering depth, and ethical considerations would strengthen its impact. The study is well-structured, and with some refinements, it could be a strong addition to the field.

Weaknesses:

  • Limited Dataset Size:

    • The dataset consists of only 500 Q&A pairs, which is relatively small for training deep learning models. While the authors acknowledge this, more discussion on how this affects model generalization would be beneficial.
    • Additional data augmentation techniques or semi-supervised learning methods could be explored.
  • Lack of Statistical Significance Analysis:

    • Performance differences between models (e.g., MARBERT vs. CAMelBERT) are not tested for statistical significance. Reporting confidence intervals or statistical tests would strengthen claims about model superiority.
  • Prompt Engineering Section Needs More Depth:

    • The study finds that few-shot learning improves GPT performance, but it does not explore different prompt formulations or optimization strategies.
    • The limited performance of GPT models (especially in zero-shot settings) suggests a need for more structured prompts or in-context learning strategies.
  • Limited Discussion on Ethical Considerations:

    • While the study mentions ethics briefly, a deeper discussion on the risks of using AI for mental health, such as biases in PLMs or potential harm from misclassifications, would be valuable.
  • Potential Overemphasis on Fine-Tuning PLMs:

    • While fine-tuning improves accuracy, it is computationally expensive. A cost-benefit analysis comparing fine-tuning with few-shot prompting (in terms of model efficiency and resource use) would be useful.

Comments, Suggestions, and Typos:

  • Abstract:

    • The phrase "Despite this potential, their efficacy in mental health applications, particularly in the Arabic language, has not been thoroughly investigated." could be made more precise—perhaps referencing specific studies or the lack thereof.
  • Section 5 (Discussion on Prompting PLMs):

    • Suggest including a more detailed explanation of why GPT-3.5 performed better than GPT-4 in the given tasks.
    • The discussion on prompt engineering would benefit from a clearer breakdown of the strategies used to optimize GPT-3.5 and GPT-4 performance. Specifically, the paper should describe how different prompt formulations were tested, including variations in wording, structure, and the use of zero-shot vs. few-shot learning. For instance, in the zero-shot setting, the model was given a direct classification instruction without examples, leading to inconsistent outputs. However, in the few-shot setting, providing 2-3 labeled examples significantly improved classification accuracy, particularly for complex queries. A detailed example, such as prompting GPT-3.5 with a structured format including labeled mental health Q&A pairs, could illustrate how the model learned from patterns and reduced misclassifications. Additionally, explaining whether hyperparameters like temperature were adjusted would provide insight into the fine-tuning process.
    • Furthermore, a direct comparison of GPT-3.5 and GPT-4 performance should be discussed, focusing on why GPT-3.5 outperformed GPT-4 in certain cases. If GPT-4 struggled with ambiguity or overgeneralized responses, an example of a misclassified question could help illustrate its weaknesses. The paper should also highlight whether GPT models showed biases toward specific categories or failed to recognize emotional support-related queries, which is crucial in mental health applications. By including these details, the study would offer a more thorough analysis of prompt-based techniques and provide valuable guidance for future research on optimizing PLMs for Arabic mental health support.

Author Response

Dear reviewer, 

Thank you for your valuable feedback and thoughtful review. We truly appreciate your time and effort in evaluating our manuscript. We have uploaded a cover letter in which we have addressed all of your comments and provided detailed responses.

Reviewer 2 Report

Comments and Suggestions for Authors

This study aims to evaluate the performance of large language models (LLMs) in classifying Questions and Answers (Q&A) within the mental health care domain.

The major concern is that the contribution of the manuscript is not consistently and systematically presented (Questions and Answers classification within the mental health care domain using PLMs).

Even in the Introduction, this goal is not explicitly stated, and the research gap is not clearly visible. Readers have to go through the entire manuscript and infer the goal and the author’s thought process to understand the study’s contribution.

Here are some suggestions for improving the manuscript:

All abbreviations should be expanded on their first use. The abstract and body of the manuscript should be treated as separate sections. For example, in the introduction, PLM should be spelled out the first time it appears. please also check for typos in manuscript.

The title highlights LLMs, but the manuscript mainly focuses on PLMs. There should be consistency to avoid misleading readers.

In the Introduction, the research objective should be stated clearly and directly before highlighting the study’s contributions. Currently, it is difficult to identify the research questions (RQ), which should be explicitly addressed in the manuscript. (classifying Questions and Answers (Q&A) within the mental health care domain)

In Section 3.3 (Experimental Design), the learning approaches used in MentalQA are not described in sufficient detail. While the manuscript provides definitions for traditional feature extraction, PLMs as feature extractors, fine-tuned PLMs, and prompt engineering, more details on their application within MentalQA are needed.

In Table 4, the authors present various models and treatment scenarios for evaluating performance. However, the rationale for selecting these specific scenarios should be explained.

The evaluation mechanism needs more detailed explanation: Different learning approaches produce different outputs—how is the evaluation performed? When is a data point considered correct? Why were micro F1-score, weighted F1-score, and Jaccard score chosen as evaluation metrics?

The case study section helps in understanding the study’s approach. However, the manuscript should also discuss the practical implications of the study. Is it worthwhile to use PLMs for Q&A classification in the mental health domain?

The manuscript should include a "Threats to Validity" or "Limitations" section.

Author Response

(The authors gave the same response as above.)

Reviewer 3 Report

Comments and Suggestions for Authors

Fine paper!

Abstract should not contain abbreviatures like Pre-trained Language Models (PLMs). This should be done in the introduction and later sections

Some considerations should be done to explain why Arabic datasets are so scarce (line 59), or a connection to lines 119-123

In Research Objectives and Contributions, authors must be more incisive. They state “The primary goal of this study is to perform extensive experiments on the recently developed dataset”. With what purpose? To identify more common questions? To elaborate an automatic answer model? To study mental problems?

Also, it is unclear what is the content of ArabicMentalQA. Authors should detail the variables that are available in that dataset.

Again, in line 101 “Decision Trees offer a clearer view of decision-making…”. What is the context? Depression detection? Other?

On page 149. I wonder if Arabic mental health system also uses International Classification of Primary Care - the codifications of the chapter P-Psychological or other mental health classification system?

On section 2.4. / 3.1 Authors should detail about how data was gattered since confidentiality issues may arise due data sensitivity. Dataset is anonymized? Patients agree with divulgation? Doctors?

Table 4. One interesting conclusion is that GPT gives more or less the same accuracy than Random (for answers), implying that  IA is not ready to classify this kind of data (zero-shot). However, lines 322-331 suggests otherwise. Please explain

Other values should be bold since they are equal to the best (some CAMeIBERT values)

Authors could compare machine learning with traditional statistical models, like logistic regression, whose interpretation is straightforward (Table 4).

Exact values labels could be inserted into figure 3 bars

Figure 4. As already stated, with zero-shot for answers type GPT is worst than random. A few lines should be inserted about this question

Figure 5. Results are expected, but there is a different slop for questions /answers. Also, higher samples sizes would be interesting to see if the slops would be similar.

5.2. Limitations and Ethical Consideration. I’ve some issues concerning the use of MentalQA dataset. Is it anonymized? Patients agreed with divulgation? Doctors are comfortable with the answers?  

References. Authors should carefully read this section and correct some typos; for example 3 uses pp. 395–398; 4 uses 46–57; in 12 the year is not in bold;

Author Response

(The authors gave the same response as above.)

Round 2

Reviewer 1 Report

Comments and Suggestions for Authors

Thank you for your thorough and thoughtful revisions in response to my previous comments. I appreciate the effort you have put into addressing the points I raised. The changes have significantly improved the clarity and overall quality of the manuscript.

I am satisfied with your responses and the corresponding revisions in the text. The manuscript is now much clearer in its presentation, and the additional explanations and justifications have addressed my concerns adequately.

I have no further major comments at this stage.